# Merging Counter-Propagation and Back-Propagation Algorithms: Overcoming the Limitations of Counter-Propagation Neural Network Models

**DOI:** 10.3390/ijms25084156

**Published:** 2024-04-09

**Authors:** Viktor Drgan, Katja Venko, Janja Sluga, Marjana Novič

**Affiliations:** 1National Institute of Chemistry, Hajdrihova 19, 1001 Ljubljana, Slovenia; viktor.drgan@ki.si (V.D.); katja.venko@ki.si (K.V.); janja.sluga@ki.si (J.S.); 2Faculty of Pharmacy, University Ljubljana, Aškerčeva Cesta 7, 1001 Ljubljana, Slovenia

**Keywords:** counter-propagation CP-ANN, back-propagation-error BPE-ANN, machine learning, cheminformatics tool, QSAR model, water solubility, drug design, acute fish toxicity, bio-concentration factor

## Abstract

Artificial neural networks (ANNs) are nowadays applied as the most efficient methods in the majority of machine learning approaches, including data-driven modeling for assessment of the toxicity of chemicals. We developed a combined neural network methodology that can be used in the scope of new approach methodologies (NAMs) assessing chemical or drug toxicity. Here, we present QSAR models for predicting the physical and biochemical properties of molecules of three different datasets: aqueous solubility, acute fish toxicity toward fat head minnow, and bio-concentration factors. A novel neural network modeling method is developed by combining two neural network algorithms, namely, the counter-propagation modeling strategy (CP-ANN) with the back-propagation-of-errors algorithm (BPE-ANN). The advantage is a short training time, robustness, and good interpretability through the initial CP-ANN part, while the extension with BPE-ANN improves the precision of predictions in the range between minimal and maximal property values of the training data, regardless of the number of neurons in both neural networks, either CP-ANN or BPE-ANN.

## 1. Introduction

Machine learning methods are widely used in the new approach methodologies (NAMs) [1], which aim to replace, in whole or in part, technologies for assessing the toxicity or activity of chemicals or drugs. We often encounter the problem of data gaps due to the lack of experimental data caused by expensive and poorly repeatable experiments or by the limitations of animal testing; in such a case, machine learning methods are of great importance for the assessment of chemical hazards and risks. Two important toxicity endpoints that are approached by in silico or machine learning methods are aquatic toxicity (LC50 toward several fish species) and bio-concentration factor. For both toxicity endpoints, there have been several attempts to develop QSAR models [2,3,4,5,6] that can be used for regulatory use, which have to follow the OEDC principles, i.e., (i) a defined endpoint, (ii) an unambiguous algorithm, (iii) a defined domain of applicability, (iv) appropriateness of the training set, (v) proper validation, and (vi) mechanistic interpretation, if possible. The OECD principles are given in the OECD Guidance Document on the Validation of (Quantitative) Structure-Activity Relationships ((Q)SAR) Models, OECD Series on Testing and Assessment, No. 69 [7].

Another important application of machine learning methods is in drug development, as they facilitate the exploration and utilization of large datasets in the search for lead compounds. The optimization of different molecular properties, known as lead optimization, is an important step in drug discovery once the most promising hit compounds have been identified. Low water solubility is often a limiting factor for further testing of promising lead compounds. A QSAR model for predicting water solubility could be used as a filtering tool at an early stage of hit-to-lead drug discovery. Water solubility (LogS) is in a reciprocal relationship with the octanol–water partition coefficient (log P, an important descriptor in QSAR models for LogS [8,9]) and thus similarly with the ability of molecules to cross the blood–brain barrier (BBB). Since water-soluble molecules typically have difficulty crossing lipid barriers, substances that are highly water-soluble may have trouble passing through the BBB. However, other factors, such as size, charge, and the presence of specific transport mechanisms, also play crucial roles in determining whether a substance can penetrate the BBB and access the brain. For these reasons, the transfer of molecules through the BBB is frequently tackled with data-driven modeling approaches, including artificial neural networks. Recently, a quantum artificial neural network approach, which resulted in a 3D-QSAR model based on the distribution of quantum mechanical molecular electrostatic potentials as the numerical descriptors, was published for the prediction of blood–brain barrier passage [10].

In the development of QSAR models, various linear or non-linear machine learning methods can be used. Best practices for QSAR modeling have been extensively described in the literature [11,12,13,14,15,16]. Artificial neural networks (ANNs) have an extensive variety of applications in a broad area of science and technology, including biology, physics, and chemistry [17]. ANNs are commonly applied machine learning methods that are particularly suitable for developing data-driven models when the mechanistic principle of the relationship between the object representation and the property is not known in advance.

After the initial escalation of applications of ANNs in various fields, nowadays, they are attracting the interest of researchers, who are developing artificial intelligence (AI) algorithms for the purpose of big data handling and exploitation [18,19,20,21,22]. In 2008, Baskin et al. [23] published a review of neural networks for building QSAR models, in which they stressed the naturalness, simplicity, and clarity of neural networks compared to many alternative machine learning approaches.

Here, a novel neural network modeling method is presented, which combines counter-propagation neural networks (CP-ANNs) [6,24,25,26,27] and back-propagation of errors artificial neural networks (BPE-ANNs) [28,29]. It takes advantage of the short training times, robustness, and good interpretability of the CP-ANN [30], while the addition of the BPE-ANN overcomes the limited number of predictions [29] afforded by the CP-ANN model. The newly developed method, which we abbreviated as the CP-BPE-ANN (counter-propagation–back-propagation artificial neural network), is tested on three different datasets, water solubility, acute fish toxicity, and bio-concentration factor. Following the idea of overcoming the drawback of the original CP-ANN models, where the number of predicted values are limited by the number of neurons, the exemplary datasets are selected, which have been modeled originally by the CP-ANN method and the resulting models have been published [6,9,31]. Implementing additional training of the original model with the BPE-ANN, having considered the trained neurons from the CP-ANN model an input for the BPE-ANN, generates a final model with no such limitation in predictions. Thus, we have shown that the newly developed combined method (CP-ANN + BPE-ANN) offers good interpolated values compared to original coarse-grain prediction.

## 2. Results and Discussion

The main shortcoming of CP-ANN models, namely, the limited number of possible model predictions, can lead to robust rather than highly precise predictions. The presented CP-BPE-ANN method was proposed after several initial attempts to establish linear interpolations between points in the response surface of the model (Grossberg layer) from the neuron weights in the Kohonen layer did not lead to improved model predictions. As known from the literature, the machine learning methods based on linear relationships cannot accurately simulate the water solubility of compounds based on their molecular descriptors [32,33,34,35]. For these reasons, we have introduced the combined CP-ANN and BPE-ANN method as described in Section 3.3.

Deriving a new model based on the counter-propagation model was desirable because the CP-ANN model can be used to evaluate the reliability of predictions using read-across based on the mapping of the compounds in the Kohonen layer, as well as to define the applicability domain of the model. The neurons that represent the building blocks of the CP-ANN model can be considered information-rich units, i.e., representative objects of input compounds. Therefore, it was expected that the performance of the CP-BPE-ANN model should be similar to that of the CP-ANN model (Stage I), which was trained with 1004 training compounds (in the case of the solubility dataset), although in Stage II, a smaller number of objects were available for training the BPE-ANN model, i.e., only the neurons (e.g., 225 or 400 in the case of the solubility dataset) of the pretrained CP-ANN model.

In the first of the three case studies presented in this paper, the merging of the CP-ANN and BPE-ANN was used for the prediction of the solubility of compounds in water. The results for the two models with all 17 descriptors and the two models with 6 and 9 preselected descriptors (see Appendix A) as obtained in our previous study [9] were evaluated. Table 1 shows the root-mean-square error (RMSE) of the predictions for CP-ANN models and the corresponding CP-BPE-ANN models. The training phase of Stage II with the BPE-ANN was performed with 5, 10, and 12 neurons in the hidden layer, and the learning rate and momentum term were 0.1 and 0.01, respectively (see Section 3.3 for a description of the methodology and Appendix A). The number of epochs used for training varied from 1000 to 500,000. The CP-BPE-ANN models that showed the best prediction performance for the test set compounds were considered optimal, and their performances are shown in Table 1. The RMSE values in bold are those for which the CP-BPE-ANN models have higher RMSE errors in the test and validation sets. Except for the model CP-ANN 3 (AqSol) and CP-ANN 6 (FHM 8), the performance of the CP-BPE-ANN models is better than that of the corresponding CP-ANN models. Here, we have to stress that in the case of predictions from the original CP-ANN models, the tested compounds that occupy the same neuron have the same predicted value (obtained in Grossberg layer of the CP-ANN), while the predictions from the CP-BPE ANN are not limited to one value for those tested compounds. The predicted values are calculated following the BPE ANN algorithm and give a different value for every unique input.

In the second example, the performance of the newly developed CP-BPE ANN method for acute fish toxicity is compared with the original CP-ANN model published by Drgan et al. [31]. As in the first example, the training of the BPE-ANN—Stage II, described in Section 3.3, was performed with 5, 10, and 12 neurons in the hidden layer, with a 0.1 and 0.01 learning rate and momentum term, and a number of epochs between 1000 and 500,000. In Table 1, the resulting performances can be found under the model labels CP-ANN 5 (FHM 2) and CP-ANN 6 (FHM 8).

The third example is taken from the publication by Drgan et al. [6], where the CP-ANN model for the bio-concentration dataset was developed. The prediction ability (RMS errors of predictions of training, test, and validation sets) of the original model is compared with the new, CP-BPE ANN model developed in this work (see Table 1), the model CP-ANN 7 (BCF). The BPE ANN architecture (number of hidden neurons and network parameters) is the same as in the first two examples.

The number of input neurons in BPE-ANN are always equal to the number of descriptors, i.e., to the dimension of input objects in each case study, while the output neuron is always just one, as we model one property (target) in each specific case study.

In QSAR modeling, the definition of the applicability domain is important for evaluating the reliability of the model predictions. *“The applicability domain of a (Q)SAR is the physico-chemical, structural, or biological space, knowledge or information on which the training set of the model has been developed, and for which it is applicable to make predictions for new compounds. The applicability domain of a (Q)SAR should be described in terms of the most relevant parameters, i.e., usually those that are descriptors of the model. Ideally the (Q)SAR should only be used to make predictions within that domain by interpolation not extrapolation.”* [36,37]. The applicability domain of CP-ANN models is commonly defined based on the distribution of Euclidean distances of the compounds in the training and/or test set [9,31,38] to the excited/central neuron. The applicability domain definition proposed by Minovski et al. [38] was used in this study. The applicability domain of the combined CP-BPE ANN models is actually tested only in the first stage (Stage I, see Section 3.3) of modeling (CP-ANN part), as the second stage (Stage II, see Section 3.3) is trained with neurons from the CP-ANN and thus the applicability domain is not altered. Here, we show the outlier analysis for the solubility dataset, while for the other two example datasets and models, the applicability domain assessment can be found in the original papers [6,31]. For the solubility models, most of the validation set compounds were in the applicability domain of the models. Table 2 lists the compounds that were outside the applicability domain and their experimental solubility compared to the predicted solubility by the CP-ANN and CP-BPE-ANN models.

The outliers (range of structural applicability) were only detected in two models, CP-BPE-ANN 1 and CP-BPE-ANN 2. One compound was common in both cases, namely, ID 330, and this compound is shown in the Appendix A. It can be seen from Table 2 that the error of the predicted value of the outliers is not always large, but the prediction should still be considered unreliable.

As discussed above, the applicability domain assessment of the CP-BPE-ANN models and the consequent outlier detection is equal to the original, already published CP-ANN models. Here, we repeated the assessment of model performance for acute fish toxicity, model CP-ANN 6 (FHM 8), as this model performs with a lower RMS of the external set of compounds than any of the newly developed CP-BPE-ANN models (CP-BPE-ANN 6-1, CP-BPE-ANN 6-2, and CP-BPE-ANN 6-3, see Table 1). Considering only the compounds within the applicability domain, the RMS errors obtained with the newly developed model fall significantly below the equivalent RMS errors of the external test compounds in the original CP-ANN models [31]. A detailed description is given in the Appendix A.

## 3. Materials and Methods

### 3.1. Datasets

#### 3.1.1. Water Solubility

A well-curated dataset for water solubility of several thousand compounds (AqSolDB) [32], which has been used for developing the original CP-ANN models [9], was used for the first example case study. Only 1674 out of over 9900 compounds were extracted, as only these passed the most severe reliability test as described by M. C. Sorkun et al. [32,33] and the strict selection criterion described in our previous work [9]. All of the compounds in the dataset contain information about their structure and properties, i.e., SMILES notations and LogS values for solubility.

The experimental solubility values (mol/L) and standardized logS units in AqSolDB [32] were obtained from aqueous solubility assays that followed the OECD guidelines for the testing of chemicals. The data were divided into three datasets: 1004 compounds in the training set TR (60%), 335 in the test set TE (20%), and 335 in the validation set V (20%). The split of compounds into datasets was based on mapping and visualization of compounds on top-map with CPANNatNIC software [6,9]. The size of the network, having 20 × 20 neurons, was small enough that several structurally similar compounds excited each neuron. This made it possible to select compounds uniformly from the entire Kohonen top-map for all three sets; see a detailed description in the reference Sluga et al. [9] SMILES representations of structures, the division of compounds into TR, TE, and V sets, and experimental values of all compounds are listed in Appendix A.

#### 3.1.2. Acute Fish Toxicity

Original data were compiled in the database by the MED-Duluth group [39] and later curated within the ANTARES project. Only 566 compounds with a definite experimental toxicity value related to 96-h LC50 for fathead minnow were selected for the initial CP-ANN model published by Drgan et al. [31]. The entire set of compounds was split into subsets using a Kohonen neural network with 20 × 20 neurons. Then, a selection of compounds were utilized to cover as much information as possible in each set. The training set contains 60% of compounds, while 40% of compounds are in the external set. A detailed description can be found in the original publication [31]. Two models from this publication are used as our second example case study.

#### 3.1.3. Bio-Concentration Factor

Bio-concentration factor (BCF) data were obtained from the work of Gissi et al. [40,41]. As described in those publications, the entire dataset of 834 chemicals was split into training, test, and validation sets, each with 606, 152, and 76 compounds, respectively. The same data were used for developing the CP-ANN model by Drgan et al. [6] and this model is used for our third example case study.

### 3.2. Molecular Descriptors

The chemical structures of all compounds were encoded into a vector representation with molecular descriptors (MDs) as vector components. In the first example case, 17 MDs, topological and physico-chemical descriptors, and the solubility values logS were obtained from AqSolDB [32]. The descriptors can be re-calculated for any new molecule with RDKit software (Version Release_2023_09_4) [42]; the descriptors are listed in Appendix A.

The descriptors of the chemical structures used in the second and third example case studies, i.e., acute fish toxicity toward fat head minnow and bio-concentration factors, were calculated using Dragon 7.0 software for molecular descriptor calculations [43]. For the acute fish toxicity dataset, one-dimensional (1D) and two-dimensional (2D) molecular descriptors were calculated. A total of 472 initially calculated descriptors were reduced to 155 by removing descriptors with high correlations (above 0.95) or low variance (below 0.005) of descriptor values. These descriptors were additionally reduced by grouping descriptors with Kohonen neural networks using transposed data matrix. From each neuron, one descriptor with a minimal and one descriptor with a maximal distance to the central neuron was selected. In this way, a pool of 89 molecular descriptors was obtained (see [31] for details).

The descriptors for the bio-concentration dataset were calculated using Dragon 7.0 software for molecular descriptor calculations. The data used can be found in the supplementary material of the original publication [6].

### 3.3. Counter-Propagation–Back-Propagation Artificial Neural Network (CP-BPE-ANN)

The CP-BPE-ANN is a two-phase network which is trained in two separated stages. In the first stage, the CP-ANN model is built, using the training dataset. This stage includes the optimization of the CP-ANN architecture and network parameters, as well as the selection of variables (in our case, these are molecular descriptors). See the Appendix A for details on the CP-ANN architecture and training algorithm. In the second stage, the BPE-ANN (see the Appendix A for details on the BPE-ANN) is trained by the neurons obtained in the trained CP-ANN model. The CP-ANN and BPE-ANN are trained separately, in this sequential order, i.e., first, the CP-ANN is trained with the compounds from the training set, and after that the neurons from the CP-ANN are extracted and used as the input training set for the BPE-ANN model. In this way, the final CP-BPE-ANN method can give different predictions for different compounds that excite the same CP-ANN neuron, overcoming the limitation of a single-value prediction per neuron in the CP-ANN models. The workflow is schematically presented in Figure 1.

The predictions are obtained by inserting the target object into the second-stage model, i.e., the BPE-ANN, which has been trained by the neurons from the previously trained CP-ANN model. Throughout the learning process, the CP-ANN model incorporates the property values of the training dataset into its output (prediction) layer, which is limited to the number of its neurons. By extending the CP-ANN training to the BPE-ANN adapted to the CP-ANN architecture, we obtain a well-covered range of predictions (any real number), regardless of the number of neurons in both neural networks, either the CP-ANN or BPE-ANN.

#### 3.3.1. CP-BPE-ANN “Stage I”

Training of the CP-ANN model. The CP-ANN was applied to build a QSAR model. It is based partially on unsupervised learning—mapping of the molecules into a Kohonen (input) layer—and on supervised learning in the Grossberg (output) layer, which represents, after it has been trained, the response surface [25]. The input data are molecules from the training dataset, represented with molecular descriptors X_is_, *i =* 1 *− n,* and *s =* 1 *− m*, for *n* descriptors and m molecules, and their target properties T_s_, *s =* 1 *− m*. Details on CP-ANN training and optimization of the network parameters are given in the Appendix A. The software for CP-ANN modeling was developed in-house, initially written in FORTRAN and compatible with MS Windows. Furthermore, it was upgraded to CPANNatNIC software, version 1.01 [6], which is based on the freely available Java environment and CDK [44] library.

#### 3.3.2. CP-BPE-ANN “Stage II”

Training of the BPE-ANN model. The BPE-ANN was used to construct a model for a set of training data obtained in “Stage I” described above, i.e., the neurons of the previously trained CP-ANN (W_ij_, *i =* 1 *− n*, and *j =* 1 *− k*, with *n* representing the number of descriptors, and k the number of neurons in the trained CP-ANN), together with associated predicted target properties P_j_, *j =* 1 *− k*. The multilayer feedforward organization of the computational units (neurons) is the most used neural network architecture and is suitable for applications in a broad area of science and technology. The multilayer feedforward architecture provides a flexible framework for modeling complex relationships in data and is widely used in diverse fields due to its effectiveness and adaptability. The neurons organized in layers (input, hidden, and output layer) receive the same information—an output vector from the previous layer—and in turn send their output vector as input to the neurons in the successive layer. The units of the input layer receive their input in the form of a data file, here represented by W_ij_, *i =* 1 *− n*, and *j =* 1 *− k* (all *k* neurons are from the trained CP-ANN). The units of the output layer produce the output signal, which is, during the training phase, compared to the target properties (P_j_, *j =* 1 *− k*) obtained from the Grossberg layer of the trained CP-ANN. Details on BPE-ANN training and optimization of the network parameters are given in the Appendix A. The MS Windows compatible software for BPE-ANN modeling written in FORTRAN was developed in the research group of the authors and is available on request.

## 4. Conclusions

The unique combination of counter-propagation and back-propagation-of-error learning algorithms for neural networks (CP-ANN and BPE-ANN, respectively) led to a new method for artificial neural networks suitable for data-driven modeling of molecular properties. The combined method, called the CP-BPE-ANN, can be further used for modeling of any other endpoint of interest. In our case, it was tested on three datasets previously used for developing CP-ANN models for water solubility, acute fish toxicity, and bio-concentration factor. The results were compared with the original CP-ANN models [6,9,31] using the same dataset. The advantage of the newly developed ANN method lies not only in a lower prediction error, but also in the fact that it overcomes the limitation of the original CP-ANN method, which allows for only predictions of property values declared in the output layer (each neuron carries only one property value). In the newly developed CP-BPE-ANN method, the property values are also interpolated with respect to the values in the neighboring neurons.

## Figures and Tables

**Figure 1 ijms-25-04156-f001:**
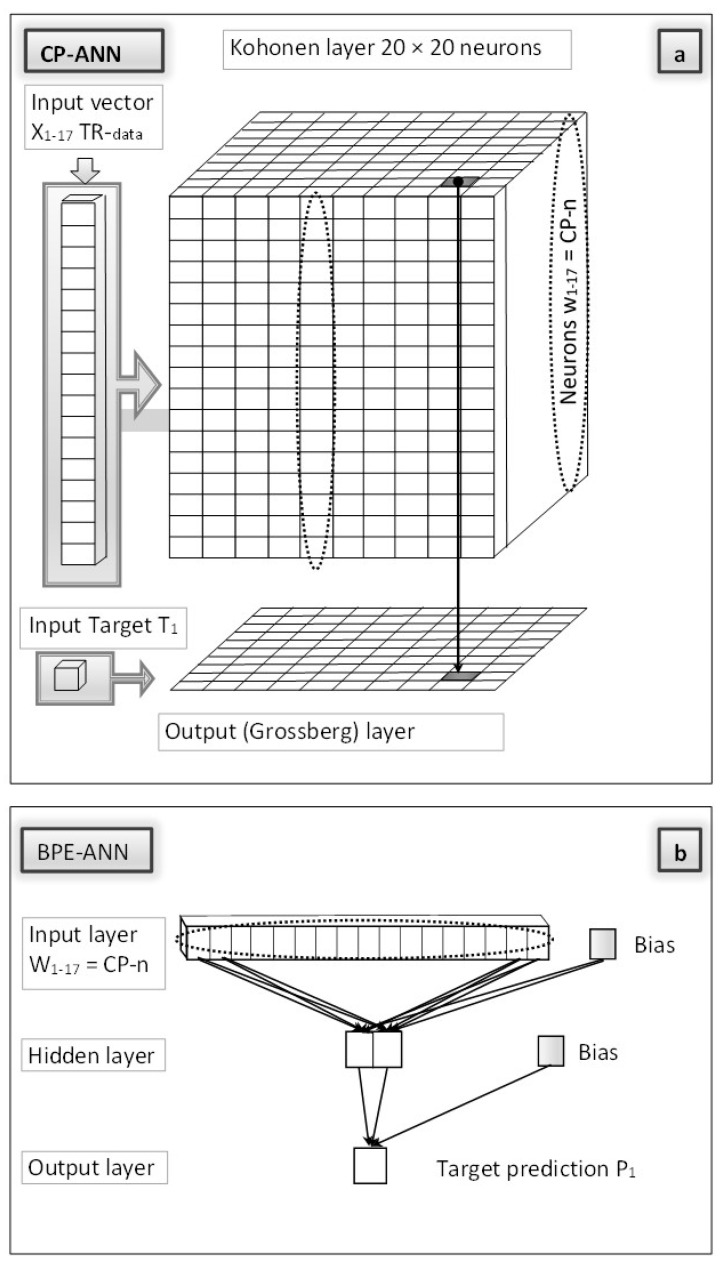
The workflow of the CP-BPE-ANN. The CP-ANN (**a**) and BPE-ANN (**b**) are trained separately, in this sequential order, i.e., first, the CP-ANN is trained with the compounds from the training set, and after that, the neurons from the CP-ANN are extracted and used as the input training set for the BPE-ANN model.

**Table 1 ijms-25-04156-t001:** Performance of CP-ANN and corresponding ^1^ CP-BPE-ANN models.

Model	Number of Descriptors	^3^ Size	^2^ RMSE			
			^4^ Training (Neurons)	Training (Compounds)	Test	Validation
CP-ANN 1 (AqSol)	17	20 × 20	/	0.68	0.88	0.92
CP-BPE-ANN 1-1		5	0.88	0.99	0.86	0.83
CP-BPE-ANN 1-2		10	0.71	0.90	0.86	0.83
CP-BPE-ANN 1-3		12	0.73	0.89	0.86	0.80
CP-ANN 2 (AqSol)	17	20 × 20	/	0.68	0.90	0.91
CP-BPE-ANN 2-1		5	0.84	0.98	0.89	0.84
CP-BPE-ANN 2-2		10	0.74	0.92	0.84	0.88
CP-BPE-ANN 2-3		12	0.74	0.92	**0.98**	0.88
CP-ANN 3 (AqSol)	6	15 × 15	/	0.79	0.80	0.80
CP-BPE-ANN 3-1		5	0.95	1.10	**0.93**	**0.90**
CP-BPE-ANN 3-2		10	0.84	1.04	**0.91**	**0.83**
CP-BPE-ANN 3-3		12	0.78	1.00	**0.89**	**0.83**
CP-ANN 4 (AqSol)	9	20 × 20	/	0.66	0.85	0.86
CP-BPE-ANN 4-1		5	0.85	0.95	0.85	0.81
CP-BPE-ANN 4-2		10	0.82	0.93	**0.93**	0.80
CP-BPE-ANN 4-3		12	0.79	0.91	**0.96**	**0.94**
CP-ANN 5 (FHM 2)	23	18 × 18	/	^5^ 0.87	/	1.01
CP-BPE-ANN 5-1		5	0.45	0.69	/	0.95
CP-BPE-ANN 5-2		10	0.38	0.65	/	0.94
CP-BPE-ANN 4-3		12	0.36	0.66	/	0.91
CP-ANN 6 (FHM 8)	28	25 × 25	/	^6^ 0.82	/	0.98
CP-BPE-ANN 6-1		5	0.23	0.70	/	**1.06**
CP-BPE-ANN 6-2		10	0.21	0.68	/	**1.06**
CP-BPE-ANN 6-3		12	0.20	0.68	/	**1.04**
CP-ANN 7 (BCF)	10	9 × 9	/	0.68	0.84	0.93
CP-BPE-ANN 7-1		5	0.30	0.70	0.78	0.79
CP-BPE-ANN 7-2		10	0.24	0.68	0.83	0.88
CP-BPE-ANN 7-3		12	0.20	0.69	0.84	0.88

^1^ Training parameters of BPE (Stage II): learning rate = 0.1, momentum term = 0.01, and number of epochs = 1000. ^2^ RMSE represents root-mean-square error of predictions for compounds in training, test, and validation sets. ^3^ Size represents the number of neurons in the counter-propagation model (9 × 9 or 15 × 15 or 18 × 18 or 20 × 20) and the number of hidden neurons in the back-propagation model. ^4^ Training objects are neurons from trained CP-ANN models. ^5^ Cross-validation of training set (leave 80 out). ^6^ Cross-validation of training set (leave 20 out).

**Table 2 ijms-25-04156-t002:** List of compounds that were outside applicability domain of four models trained with CP-ANN method. Predictions of both CP-ANN and CP-BPE-ANN models are given.

Model	ID_outlier_		^1^ LogS	
		^2^ Exp.	^3^ Pred_CP-ANN_	^4^ Pred_CP-BPE-ANN_
CP-BPE-ANN 1	330	−5.418	−4.114	−5.653
CP-BPE-ANN 2	330	−5.418	0.057	−0.460
	638	−6.294	−6.134	−6.052
CP-BPE-ANN 3	/	/	/	/
CP-BPE-ANN 4	/	/	/	/

^1^ logS units, S [mol/L]; ^2^ experimental; ^3^ predicted by CP-ANN model; ^4^ predicted by combined CP-BPE-ANN model.

## Data Availability

All data are available upon request. Software can be found at https://www.ki.si/fileadmin/user_upload/datoteke-D01/L03/SOM_tool_for_PROSIL_project.zip. accessed on 4 April 2024.

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
