# Peer review of "Merging Counter-Propagation and Back-Propagation Algorithms: Overcoming the Limitations of Counter-Propagation Neural Network Models"

_ijms, 2024, doi:10.3390/ijms25084156_

Round 1

Reviewer 1 Report

Comments and Suggestions for Authors

Overall this is a nice paper that describes several large models developed on very large databases.  Key features that describe the models are not included, such as outliers and broad areas of validation.  Overall - it is an excellent contribution that is in need of minor revisions.

The introduction covers the field, but could be expanded.  The experimental section describes the work done so that it can be repeated.  The results are presented, but more specifics could be provided.  The discussion can be expanded.  The conclusions support the results.

Below are specific comments that may improve the paper depending how the authors address these comments:

1)  Water solubility is related to the  blood brain barrier.  The authors could discuss their results in light of a recent publication related to the blood brain barrier using QSAR and Neural Networks.

Kim, T.; You, B.H.; Han, S.; Shin, H.C.; Chung, K.-C.; Park, H. Quantum Artificial Neural Network Approach to Derive a Highly Predictive 3D-QSAR Model for Blood–Brain Barrier Passage. Int. J. Mol. Sci. 2021, 22, 10995. https://doi.org/10.3390/ijms222010995

2) The authors could explain how traditional descriptors like log P are related to water solubility, and why their neural network QSAR is better.

3) How does this method compare to implicit solvation models, such as COSMO-RS or other solvation models that are empirical and parameter based.

4) The results are presented with strong statistical validity.  The discussion of the validity of the models could be expanded.

 5) Where can the software be downloaded by others to use?

 6) The dataset considered is impressive – just less than 10,000 compounds.  How many outliers were found for the models and why are these compound outliers.

 7)  The number of descriptors in the model are huge (over 15).  How can these models have scientific meaning?   It would be better if the models had fewer descriptors that have scientific value and meaning behind the models.

8)  With such a large number of compounds and models, there is a high probability of overfitting and chance correlation.  The authors could expand on the methods and procedures that were used to prevent overfitting and chance correlation.  A simple solution is to present the results for models obtained with 5 or fewer descriptors and then expand the discussion to include the value of the additional descriptors.

9)  Table 1 should appear on one page.

10)  Table 2 should appear on one page

11) The validation parameters for the equations could appear in an addition table.  This includes leave one out cross validation, and other validation parameters (R2 of a test set etc).

Author Response

Dear Reviewer,

Reviewer 2 Report

Comments and Suggestions for Authors

Dear Authors,

Couple of thoughts and my suggestion below.

1. The paper is very concise. This is normally fine, but here I have an impression that reader would appreciate some high level information about key points of the paper. E.g. the strong reduction of the AqSolDB from over 9900 to only 1674. Please elaborate more on that.

2. In line 46 you refer to some review of application of NNs to QSAR problems. Well, 16 years that passed since that review was published is like an epoch, in between there were plenty of other analogous reviews published and these should be analysed and referenced. In general introduction is very concise and requires significant extension.

3. Could you please outline in the paper what are key differences between your approach and standard fully-connected layers? It is critical to thoroughly compare presented approach to existing methods, such that the reader can understand what is the novelty.  

4. Please elaborate about the connection between the CP-ANN and BPE-ANN components. Are they trained simultaneously? Due to the fact that the description is so concise I have difficulties in grasping key points.

5. The entire Appendinx B sounds like a standard backpropagation algorithm description? Is that correct? If so, I would recommend removing this from the manuscript.

6. The performance of the method is not compared to the other methods, there are plenty of aq. solubility QSAR models published. It's critical to thoroughly compare your method to these published results. 

7. You have excluded a significant fraction of the instances from the AqSolDB set. Why not to treat the excluded data as an additional test set?

8. The link to the software (https://www.ki.si/en/departments/d01-theory-department/laboratory-for-cheminformatics/soft-235) does not work.

I could find much more areas for improvement. In summary: I cannont recommend the paper to the publication in IJMS.

Kind regards,

Reviewer

Author Response

Dear Reviewer,

Reviewer 3 Report

Comments and Suggestions for Authors

A very technical revisiting of the solubility prediction problem, showing that one NN architecture is better than another on a good old benchmarking set. Nice, but the goal is to... predict solubility, not to glorify a particular ML method.

The keys to successful machine learning are, in order of importance (a) the representativity of the sample of training items for the problem space [what is not taught cannot be learned!], (b) the quality of descriptors and (c) the ML algorithm. It makes no sense at all to make generic claims that ML algorithm A is better than algorithm B - this may be true with a given descriptor set, and false with another descriptor set. In particular, if you'd include... calculated logS as a descriptor, and if the model beyond that descriptor would have been based on your actual training set... then linear regression would be the most sophisticated algorithm to use (with slope 1.0 and intercept 0.0, if you don't mind). You only need NNs because you took the (cheap) solution of downloading the 17 (basic) descriptors from the solubility database, rather than invest effort to hunt down or even design specific molecular descriptors which would capture solubility well. So, all your incremental progress in fitting the new NN may be washed away by the discovery of new and better descriptors that might make solubility prediction much easier! Therefore, the strategic decision you have to take is the following: do you want to develop solubility models - or do you want to develop new NN architectures? Both are nice goals to pursue - but not in the way you did it. If it's solubility you're after, you'll conduct a global research for optimal descriptors in combination with the ML technique working best with them. If you want to make the NN architectures progress, then you'll need to benchmark them not against one, but against tens or hundreds of alternative prediction problems, before you can claim that one architecture has systematic advantage over another. What you did here is neither, nor... so it has no real interest so far. It MIGHT be important and novel, but that is not proven yet.

Comments on the Quality of English Language

Globally, Engish is fine in spite of some rather funny expressions like molecules being "equipped with SMILES notations and solubility logS values". So are molecules equipped with SMILES strings more expensive than cars equipped with a GPS system?

Author Response

Dear Reviewer,

Round 2

Reviewer 3 Report

Comments and Suggestions for Authors

Thank you for your detailed reply - indeed, this is much clearer now. We're not yet done with benchmarking but - so far, so good - this paper reports an improvement of NN algorithms and can be published as such

Comments on the Quality of English Language

Fine with me